# First-time comparison between NO₂ vertical columns from GEMS and Pandora measurements

Serin Kim[1], Daewon Kim[1], Hyunkee Hong[2], Lim-Seok Chang[2], Hanlim Lee[1], Deok-Rae Kim[2], Donghee Kim[2], Jeong-Ah Yu[2], Dongwon Lee[2], Ukkyo Jeong[1], Chang-Kuen Song[3], Sang-Woo Kim[4], Sang Seo Park[3], Jhoon Kim[5], Thomas F. Hanisco[6], Junsung Park[1], Wonei Choi[1], Kwangyul Lee[7]

[1]Division of Earth Environmental System Science, Major of Spatial Information Engineering, Pukyong National University, Busan, Republic of Korea
[2]Environmental Satellite Center, National Institute of Environmental Research, Incheon, Republic of Korea
[3]Department of Urban & Environmental Engineering, Ulsan National Institute of Science and Technology, Ulsan, Republic of Korea
[4]School of Earth and Environmental Sciences, Seoul National University, Seoul, Republic of Korea
[5]Department of Atmospheric Sciences, Yonsei University, Seoul, Republic of Korea
[6]Atmospheric Chemistry and Dynamics Lab, NASA Goddard Space Flight Center, Greenbelt, MD, USA
[7]Air Quality Research Division, Climate and Air Quality Research Department, National Institute of Environmental Research, Incheon, Republic of Korea

*Correspondence to*: Daewon Kim (k.daewon91@gmail.com)

**Abstract.** The Geostationary Environmental Monitoring Spectrometer (GEMS) is a UV–visible spectrometer onboard the GEO-KOMPSAT-2B satellite launched into a geostationary orbit in February 2020. To evaluate the GEMS NO₂ total column data, a comparison was carried out using the NO₂ vertical column density (VCD) measured direct sunlight using the Pandora spectrometer system at four sites in Seosan, South Korea, from November 2020 to January 2021. Correlation coefficients between GEMS and Pandora NO₂ data at four sites ranged from 0.35 to 0.48, with root mean square errors (RMSEs) from $4.7 \times 10^{15}$ molec. cm⁻² to $5.5 \times 10^{15}$ molec. cm⁻² for cloud fraction (CF) < 0.7. Higher correlation coefficients of 0.62–0.78 with lower RMSEs from $3.3 \times 10^{15}$ molec. cm⁻² to $4.3 \times 10^{15}$ molec. cm⁻² were found with CF < 0.3, indicating the higher sensitivity of GEMS to atmospheric NO₂ in less-cloudy conditions. Overall, the GEMS NO₂ total column data tended to be lower than those of Pandora owing to differences in representative spatial coverage, with a large negative bias under high-CF conditions. With a correction for horizontal representativeness in Pandora measurement coverage, correlation coefficients ranging from 0.69 to 0.81 with RMSEs from $3.2 \times 10^{15}$ molec. cm⁻² to $4.9 \times 10^{15}$ molec. cm⁻² were achieved for CF < 0.3, showing better correlation with the correction than without the correction.

## 1 Introduction

Nitrogen dioxide (NO₂) is a key species in the troposphere and stratosphere for atmospheric chemistry and air quality (Crutzen, 1979; Seinfeld and Pandis, 1998), and is mainly emitted by anthropogenic sources, such as fossil fuel combustion in vehicles and power plants. Natural sources, such as lightning, biomass burning, and soil microbial action are also major contributors to

atmospheric $NO_2$ (Crutzen, 1979). $NO_2$ is a precursor of tropospheric ozone, aerosols, and hydroxyl radical (OH) (Boersma et

al., 2009), and high concentrations affect the lifetime of atmospheric $CH_4$ and direct radiative forcing of the atmosphere (Pinardi et al., 2020). In addition, the $NO_2$ diurnal cycles are important factors for understanding temporal patterns such as NOx emissions, chemistry, deposition, advection, diffusion, and convection (Li et al., 2021).

Therefore, it is important to monitor $NO_2$, and representative methods for this are as follows. Chemiluminescence-based in-situ instruments have provided a highly accurate $NO_2$ mixing ratio at a measurement location, but with limited spatial coverage

(e.g., Bechle et al., 2013; Jeong and Hong, 2021). Satellite-based remote sensing instruments on polar orbits, such as the GOME-1/2 (Global Ozone Monitoring Experiment; Burrows et al., 1999; Munro et al., 2016), SCIAMACHY (Scanning Imaging Spectrometer for Atmospheric Cartography; Bovensmann et al., 1999), OMI (Ozone Monitoring Experiment; Levelt et al., 2006), and TROPOMI (TROPOspheric Monitoring Instrument; Veefkind et al. 2012), have effectively complemented the ground-based observations by providing global distribution of $NO_2$ total column density (Lamsal et al., 2014). The recently

GEMS (Geostationary Environment Monitoring Spectrometer; Kim et al., 2020) onboard the GEO-KOMPSAT-2B (Geostationary Korea Multi-Purpose Satellite 2B) was launched in February 2020. The NIER (National Institute of Environment Research), where the GEMS ground station is operated, has been transmitting the GEMS products including $NO_2$ Vertical column density (VCD) in real time from December 2022. GEMS Map of the Air Pollution (GMAP) campaigns have taken place from 2020 and are also scheduled to be held annually to evaluate the quality of the GEMS measurements of trace

gas and aerosol products based on trace gases, aerosol composition and optical property measurements at various platforms. This study conducted the first quick evaluation via comparison between the GEMS $NO_2$ VCDs and those of Pandora measurements at several sites in a suburban area in Korea during the first GMAP campaign in 2020 winter. We evaluate the differences between $NO_2$ VCD obtained from Pandora and GEMS especially depending on cloudy and clear sky conditions.

The comparison and validation of satellite-based $NO_2$ VCD retrievals are essential because of their non-negligible error sources

such as assumed atmospheric profiles, surface reflectance, and measurement uncertainties (Hong et al., 2017). In addition, $NO_2$ VCD retrievals from GEMS require precise assessments because the observation geometries of the geostationary Earth orbit (GEO) are different from those of the low earth orbits (LEO) and other systematic uncertainties may affect the retrievals (e.g., diurnal variations of the atmospheric profiles, which are used for air mass factor (AMF) calculations). Ground-based remote sensing instruments such as the MAX-DOAS (multi-axis differential optical absorption spectroscopy; Honninger et al.,

2004) measure scattered sunlight at various elevation angles to derive tropospheric column amounts of $NO_2$ as well as profile estimates (e.g., Irie et al., 2008; Wagner et al., 2011; Wang et al., 2017). Direct-Sun instruments such as the Pandora (Herman et al., 2009) measure direct sunlight to retrieve the $NO_2$ VCD, of which the absorption light path of the photons reaching to their detector may be shorter than those of the MAX-DOAS instruments; thus, they are less sensitive to the surface mixing ratio of $NO_2$. However, uncertainties in $NO_2$ VCD retrievals by AMF calculation are low as they use simple geometric AMF

(Herman et al., 2009). Numerous studies have utilized the recently expanding Pandonia Global Network (PGN; https://www.pandonia-global-network.org/) for validation of the polar-orbiting satellite products (e.g., Herman et al., 2009; Tzortziou et al., 2014, 2015; Herman et al., 2019; Judd et al., 2019, 2020; Pinardi et al., 2020; Verhoelst et al., 2021).

This study represents the first attempt to compare and validate NO$_2$ VCD retrievals from GEMS with Pandora instruments deployed during the GMAP (GEMS Map of Air Pollution; from November 2020 to January 2021) campaign in Seosan, South

Korea. The measurement periods and locations of the four Pandora instruments are summarized in Fig. 1 and Table 1. In Section 2, the campaign and GEMS data are explained, followed by the Pandora instrument and retrieval methodology. Section 3 provides a comparison between the instruments and between Pandora and GEMS. The results are described in three parts in Section 4: intercomparison between Pandora instruments, comparison with GEMS NO$_2$, and consideration of horizontal representativeness. Finally, the conclusions are provided in Section 5.


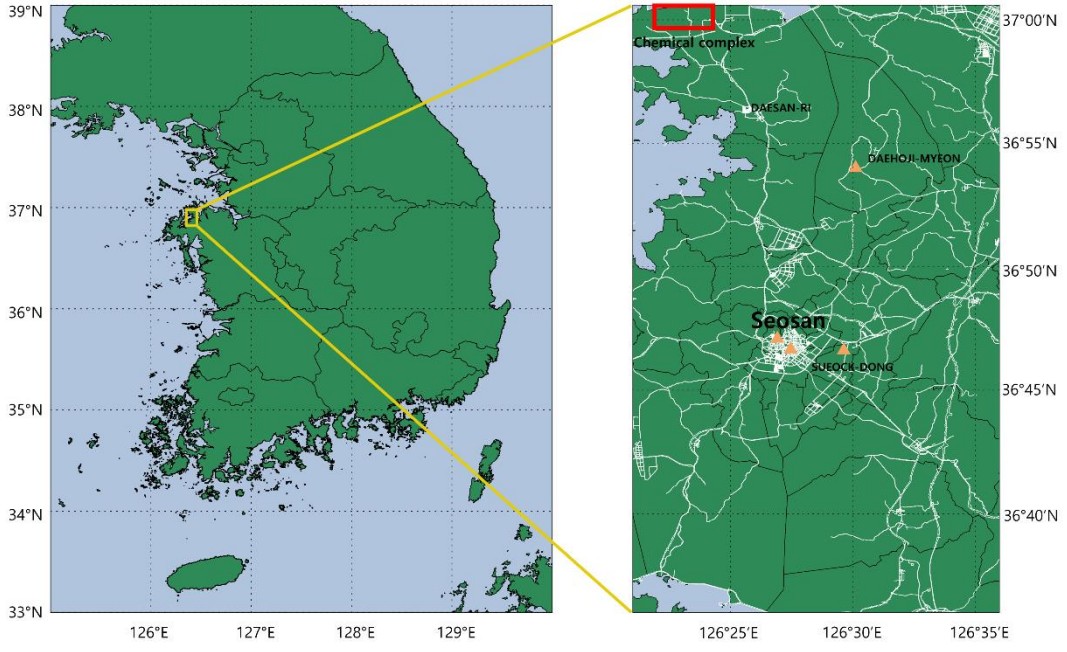

**Figure 1.** Measurement sites for the GMAP 2020 campaign. Triangles indicate observation sites

**Table 1.** The measurement sites and period.

|  | Latitude | Longitude | Period |
|---|---|---|---|
| Seosan (SS) | 36.78° N | 126.49° E | 2020.11.12–2020.12.03<br>2020.12.03–2021.01.27 |
| Seosan-CC (CC) | 36.78° N | 126.45° E | 2020.12.09–2021.01.31 |

| Daehoji (DHJ) | 36.90° N | 126.50° E | 2020.12.09–2021.01.17 |
|---|---|---|---|
| Dongmoon-2dong (DM2) | 36.78° N | 126.46° E | 2020.12.09 – 2021.01.03 |

## 2 GMAP campaign

### 2.1 The first GMAP campaign

The first GEMS validation campaign, GMAP 2020, was conducted between November 2020 and January 2021 in Seosan. The Pandora instruments used in the campaign were the standard versions described in Section. 3.1. The mean $NO_2$ concentration in Seosan for 2016–2020 was 0.017 ppm, ~0.16 % lower than the Korean national five-year average (https://www.airkorea.or.kr/web, last access: 07 March 2021). Direct sunlight measurements were conducted at four sites, as described in Table 1 and Fig.1: Seosan (SS), Seosan City Council (CC), Dongmun-2dong (DM2), and Daehoji (DHJ). Emissions from vehicular and point sources may have contributed to variations in $NO_2$ concentrations in the Pandora lines of sight, depending on the wind direction. Major roads and an agricultural complex were located within ~0.7 km of the SS site, a road and roundabout were near the CC site, a road was near the DM2 site, and a petrochemical complex was located approximately 16 km NW of the DHJ site. To estimate the differences in the $NO_2$ VCD among the Pandora instruments, an initial intercomparison was conducted for two weeks at the SS site. It should be noted that the Pandora instruments were manufactured with the same optics and spectrograph. However, it is still important to quantify the differences between the $NO_2$ columns retrieved from the four Pandoras at the same location before comparing them with the GEMS $NO_2$. Instruments were installed at these four sites to measure direct sunlight from December 2020 to January 2021. The measurement periods varied according to the instrument conditions (Table 1).

### 2.2 GEMS $NO_2$ data

GEMS, a hyperspectral UV-Vis image spectrometer covers a wavelength range of 300–500 nm with a full width at half maximum (FWHM) of approximately 0.6 nm. GEMS measures atmospheric concentrations of species that affect air quality, such as $NO_2$, $O_3$, $SO_2$, HCHO, and aerosols on an hourly basis from 00:45 to 05:45 UTC with a spatial resolution of $3.5 \times 8$ km (Kim et al., 2020). The GEMS $NO_2$ column retrieval was based on the DOAS algorithm (Platt and Stutz, 2008) at wavelength intervals of 432–450 nm (Park et al., 2020). The GEMS cloud fraction (CF) is retrieved using $O_2$–$O_2$ absorption properties and DOAS (Choi et al., 2020). We used CF for the comparison of $NO_2$ VCDs (more details, see Sect. 3). For data evaluation, we used GEMS L2 $NO_2$ VCD version 1.0, which was available immediately after the IOT (in Orbit Test) carried out in July 2020.

## 2.3 Pandora Instrument and Spectral Fitting

Pandora is a ground-based spectrometer that measures direct sunlight over a wavelength range of 280–525 nm with an FWHM of approximately 0.6 nm. The charge-coupled device (CCD) detector in the Pandora spectrometer has 2048×64 pixels. The spectrometer is connected to a telescope "head sensor" consisting of a collimator and filters such as UV340 filter, neutral density filters, and opaque filter through an optical fiber with a 400 µm core diameter. A target area can be observed with a field of view (FOV) of up to 1.6° (Herman et al., 2018).

The four instruments used here are referred to as P1, P2, P3, and P4. The measured spectra were analyzed to retrieve $NO_2$ slant column densities (SCD) using QDOAS software (Fayt et al., 2011) based on the DOAS technique which can retrieve trace gas concentrations by separating trace gas absorption cross-section into slowly and rapidly varying parts (Honninger et al., 2004). The reference spectrum used for fitting was measured at around noon on a clear day (Herman et al., 2009). This refers to the spectrum with the lowest $NO_2$ concentration used to perform optical density fitting over a period of time. During the intercomparison, the radiance obtained at noon on November 28 (a clear day) was used as the reference spectrum for P1, P3, and P4. November 14 was used as a reference for P2 due to the lack of data on November 28th, 2020. As the $NO_2$ differential VCD (dVCD) from P2 was retrieved using a different reference spectrum, it was considered secondary data. The $NO_2$ differential slant column density (dSCD) was obtained using the absorption cross-sections for $NO_2$ (254.5K) calculated using 220K and 294K (Vandaele et al., 1998) and $O_3$ (225K) (Serdyuchenko et al., 2014), as a fourth-order polynomial in the fitting window of 400–440 nm. The wavelength range and absorption cross-section were the same as those used in PGN (https://pandora.gsfc.nasa.gov/, last access: 28 March 2022). Additionally, we used $O_4$ at 293K (Thalman and Volkamer, 2013) for the spectral fitting (see Fig. 2). This reduced retrieval error by about 0.2 %. Figure 2 shows an example of the P1 spectrum fitting results at 10:43 Local Time (LT) on November 28, 2020. The $NO_2$ VCD was obtained by dividing the $NO_2$ SCDs by the geometric AMFs. After the initial intercomparison, the reference spectrum was selected when the weather was clear with no air pollution because the instrument locations were different. P1 and P4 used the noon spectrum on January 14, 2021, as a reference spectrum, whereas P2 and P3 used spectra from December 19, 2020.

130

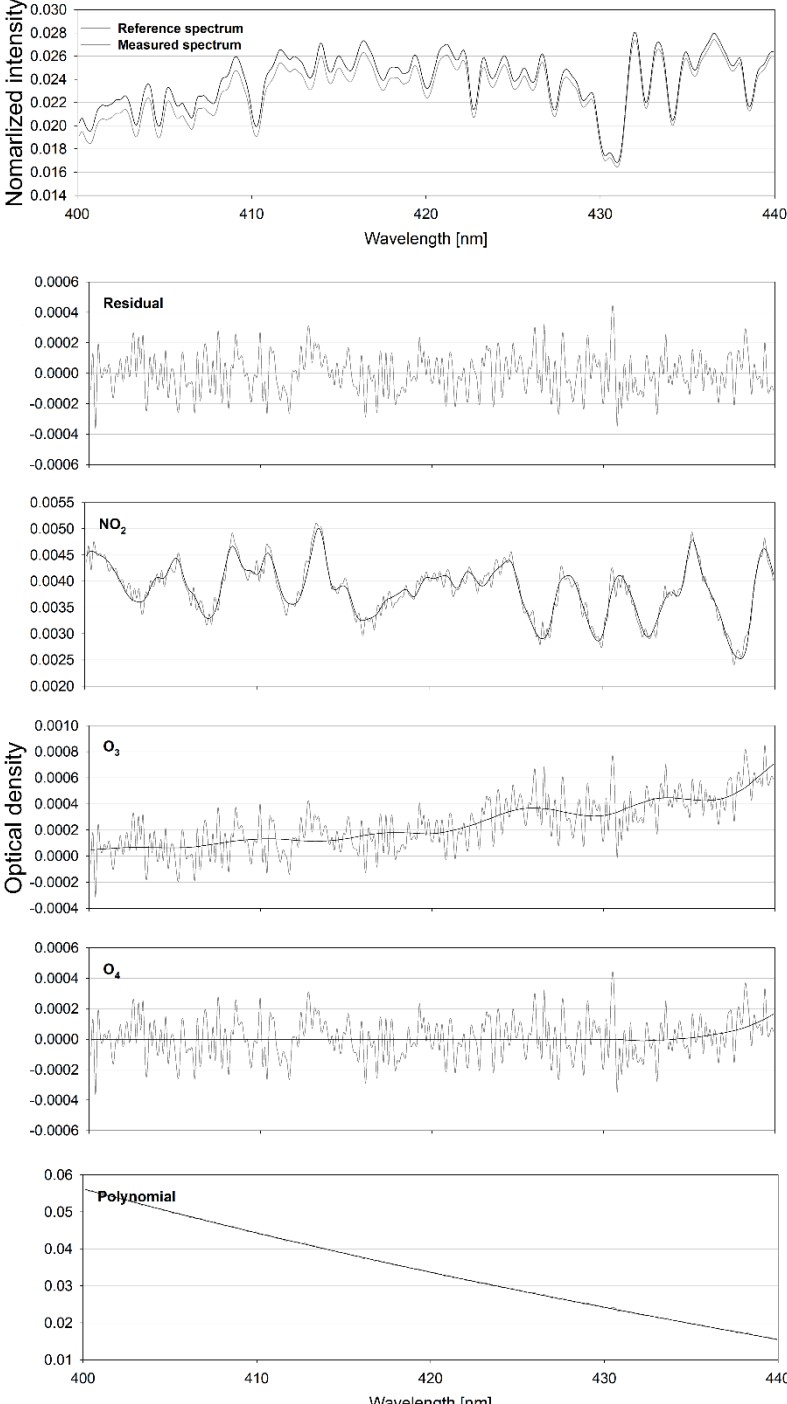

**Figure 2.** Fitted slant column optical depths example for November 28 2020 at 10:43:37 LT for P1. The black line represents the absorption signal, and the grey line represents the absorption signal and fit residual.

## 3 Method

This study aimed to evaluate the GEMS $NO_2$ VCD via quick comparisons between the GEMS $NO_2$ column data and those of
Pandora data. The differences between the Pandora and GEMS $NO_2$ data can be attributed to uncertainties in the Pandora and
GEMS $NO_2$ columns and differences in the measurement geometries. The spatiotemporal differences between the Pandora and
GEMS measurements also cause differences between the $NO_2$ column data obtained from the two platforms. To quantify the
differences in the Pandora $NO_2$ measurements, all four Pandoras performed identical direct sun measurements at the SS site
during the intercomparison period by setting the same observation schedules for all instruments. The $NO_2$ retrievals from the
four collocated Pandora instruments showed consistency of the processed data as shown in Fig. 3 and 4.  The specifications
and retrieval methods for Pandora are described in Sect. 2.3. During the intercomparison, because clear days were not sufficient
to calculate the background concentration, we compared the Pandora instruments using dVCD.  On the other hand, in the
comparison with GEMS $NO_2$, $NO_2$ VCDs from Pandora were used. As it measures direct sunlight, it is negligibly affected by
scattered sunlight. However, under cloudy conditions, all Pandora may not see the same location of the sun because of the
inhomogeneity of cloud thinness. In thick cloudy conditions compared with clear sky conditions, it may lead to the inclusion
of unwanted stray light and increase detector noise. To understand the influence of clouds, Pandora was investigated using
GEMS cloud fraction (CF) to determine whether the signal was affected by clouds.

## 4 Results

### 4.1 The intercomparison of $NO_2$ dVCD from Pandora

Pandora intercomparison was carried out from November 12 to December 3, 2020, at the SS site to quantify $NO_2$ dVCD
retrievals from the Pandora instruments. We defined dVCD as differential SCD-divided AMF with no background correction.

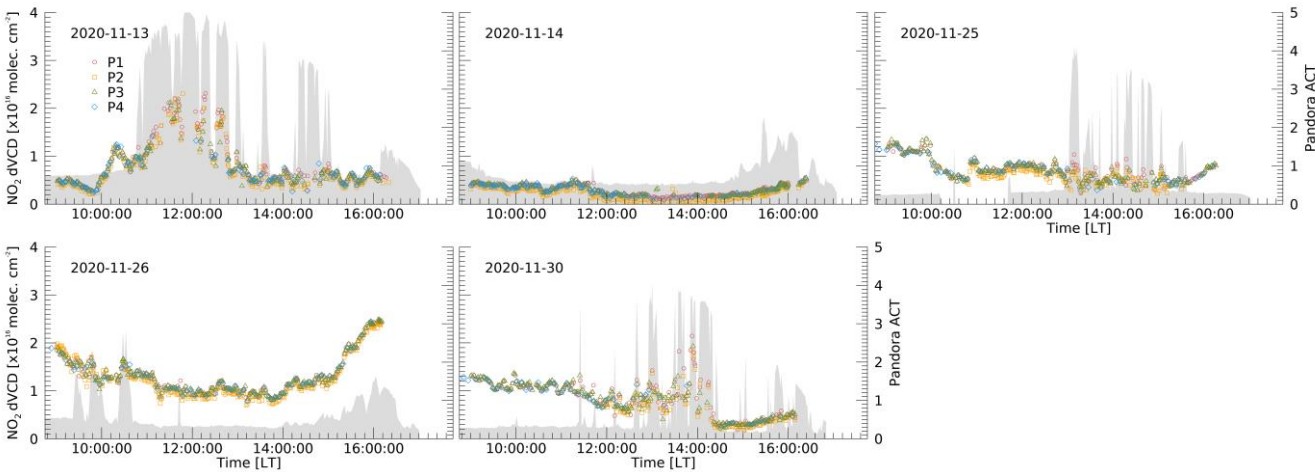

**Figure 3.** Time series of Pandora retrievals during the intercomparison. Circle (red), square (orange), triangle (green) and diamond (blue) symbols represent total NO$_2$ dVCD for P1, P2, P3, and P4, respectively. The grey shade represents Pandora aerosol cloud thickness.

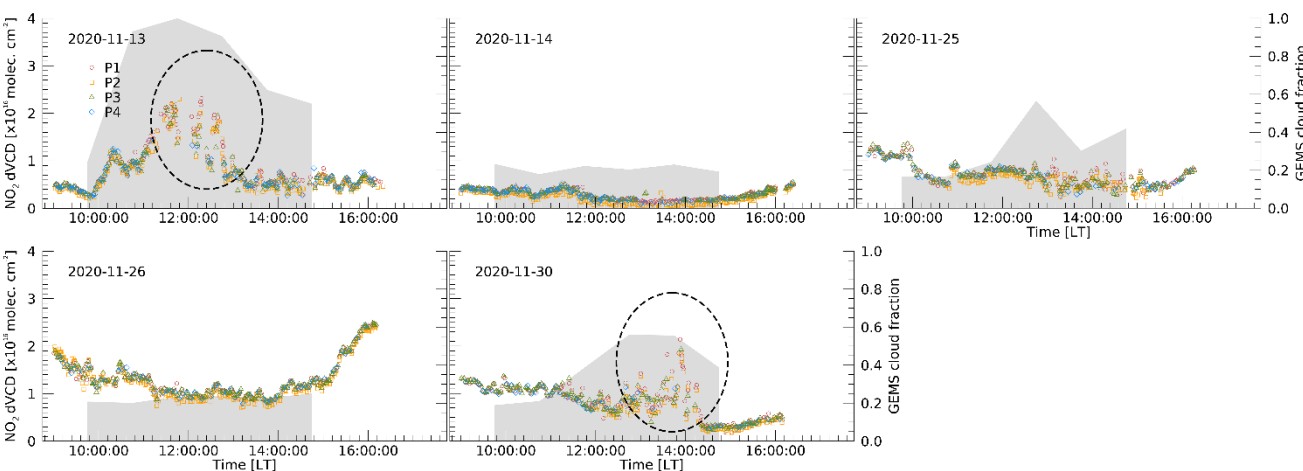

**Figure 4.** Time series of Pandora retrievals during the intercomparison. Circle (red), square (orange), triangle (green) and diamond (blue) symbols represent total NO$_2$ dVCD for P1, P2, P3, and P4, respectively. The grey shade represents the GEMS cloud fraction.

The time series of data from all instruments for the intercomparison period are shown in Fig. 3 and 4, except for the rainy days. Circles, squares, triangles, and diamond symbols represent the NO$_2$ dVCD retrieved by P1, P2, P3, and P4, respectively. The grey area in Fig. 3 represents the Pandora aerosol cloud thickness (ACT), which indicates the Aerosol Optical depth (AOD) before cloud screening. ACT was retrieved with the Spectral Measurements for Atmospheric Radiative Transfer spectroradiometer (SMART-s) algorithm developed for aerosol retrieval using the optimal estimation method (OEM) (Jeong et al., 2020). The diurnal patterns of NO$_2$ for each Pandora instrument showed good agreement. The NO$_2$ dVCD during the

period ranged from $1.63 \times 10^{14}$ molec. cm$^{-2}$ to $2.49 \times 10^{16}$ molec. cm$^{-2}$, and tend to increase during the morning and late afternoon (after 16:00). At midday, emissions are relatively lower than those during rush hour that have NO$_2$ emissions from

vehicles (Zhao et al., 2020). As Seosan is a sub-urban area, it can be affected by commuting time. As shown in Fig. 3, although there was a good agreement between the instruments, discrepancies occurred in some cases. This occurs when there are many clouds with ACT greater than about 2.5. It is considered that clouds contributed to the discrepancies, which shows certain cloud effects on the NO$_2$ retrievals from the ground-based direct sun measurements. Thus, aerosols and clouds can affect the retrieval accuracy of trace gases. Therefore, when comparing with GEMS, GEMS CF was used to consider the effects of

clouds. Before comparison with GEMS, GEMS CF was also applied during the intercomparison, and can be seen in Fig. 4. The grey area in Fig. 4 represents the GEMS CF of the GEMS observation time. The dashed-line ovals (Fig. 4) indicate periods with discrepancies between the Pandora instruments during the afternoons of November 13 and 30, similar to the case of the ACT retrieved from Pandora measurements. Although the temporal trends of ACT and GEMS CF were similar, there is a difference in spatiotemporal resolution. The GEMS spatial resolution is $3.5 \times 8$ km$_2$, and the measurement area of Pandora

could be a clear sky even if GEMS retrieved high CF. These differences sometimes result in less spread of Pandora NO$_2$ for CF $> 0.3$. Thus, we compared NO$_2$ VCDs from Pandora and those from GEMS depending on the CF conditions less than 0.3, 0.5, and 0.7. Figure 5 shows the linear regression of the NO$_2$ dVCDs from P2, P3, and P4 against those from P1, which produced the smallest fitting errors on average during the intercomparison period.

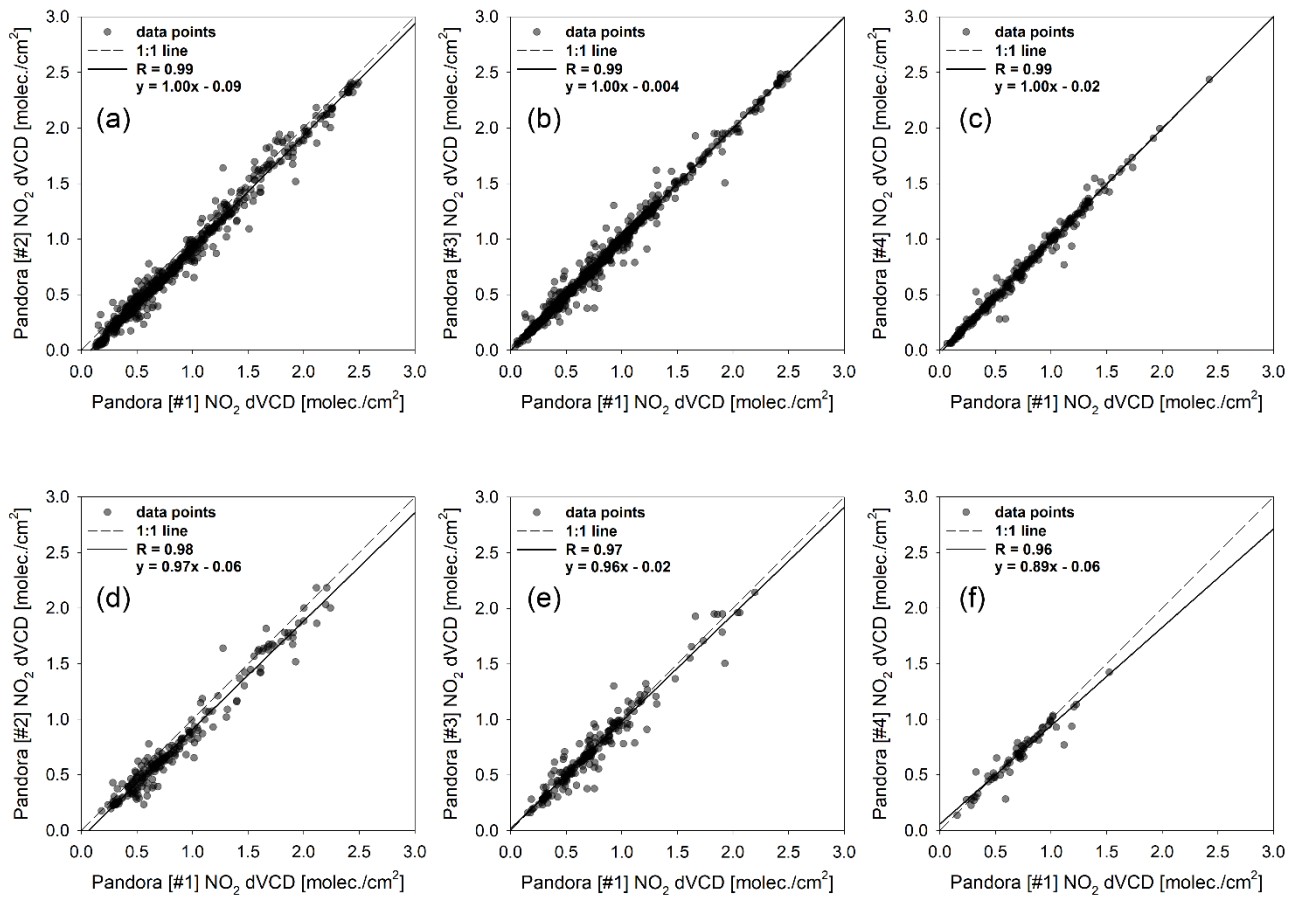

**Figure 5.** The scatter plots between P1 and others. (a), (b) and (c) shows the comparison with all data of P2, P3, and P4. (d), (e) and (f) shows comparison with P2, P3 and P4 when GEMS CF > 0.3.

In Figure 5 a, b, and c, the correlation coefficients were found to be 0.99 with a slope of 1 and an interceptor between 0.004 and 0.09, showing good agreement for all CF conditions. Overall, the $NO_2$ retrieved by each instrument yielded similar correlations, even with CF > 0.3, although the R values were slightly lower in Fig. 5 d–f, with slopes deviating further from the 1:1 line.

## 4.2 Comparison of NO₂ VCD between Pandora and GEMS

After the intercomparison period, the Pandora instruments were moved to the four sites for the observation of direct sunlight to evaluate $NO_2$ VCD for comparison with GEMS data. Measurement was carried out from December 9, 2020, and it was either snowing or raining for more than half of the measurement period. For the validation of GEMS, Pandora data were averaged within ±10 minutes from the center of the GEMS observation time. The GEMS measurement pixels are not fixed but

rather change as a function of time. Therefore, comparisons were made using the GEMS pixels closest to each Pandora station.
Comparisons were carried out between the NO₂ VCDs obtained from Pandora and GEMS at CFs of 0.3, 0.5, and 0.7. The

direct-sun DOAS (DS-DOAS) horizontal absorption path lengths are generally within 4 km with a solar zenith angle (SZA) <
50° (Herman et al., 2009). However, most SZAs were greater than 50° during the campaign period. Thus, a single GEMS pixel
may not cover the absorption path of the Pandora observations. This horizontal discrepancy was partly considered in the
comparison between the Pandora NO₂ data and those of the GEMS, which can be found in Section 4.3.

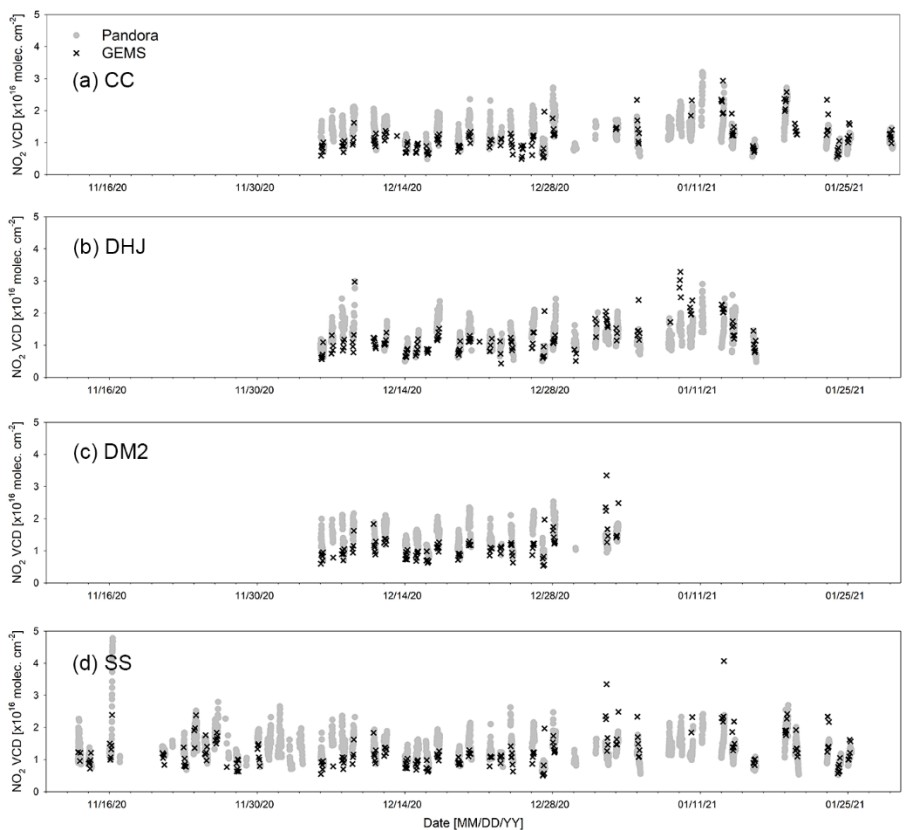


**Figure 6.** Hourly variations in NO₂ VCD were obtained from Pandora (grey full circles) and GEMS (black **x**). (a), (b), (c), and (d) represent
the CC, DHJ, DM2, and SS sites, respectively.

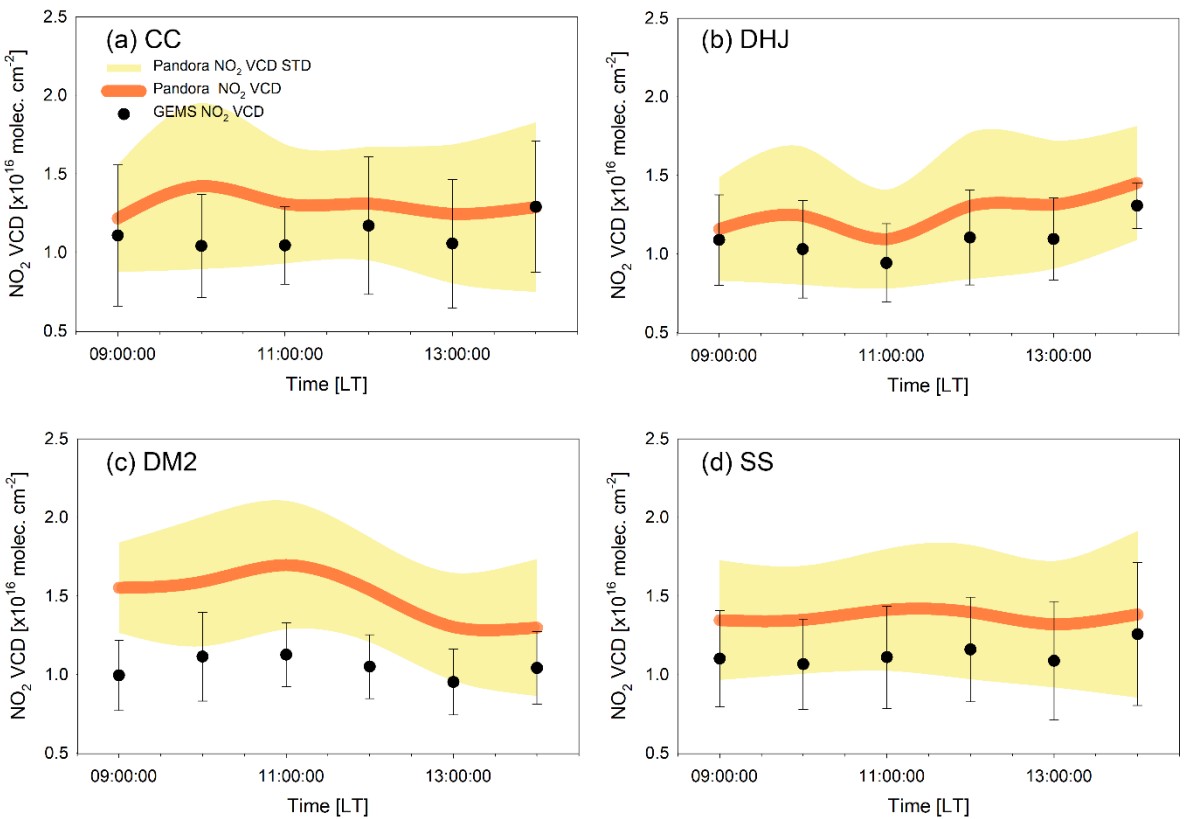

**Figure 7.** Hourly mean NO₂ VCD using only matched data from Pandora (orange line) and GEMS (black solid circles). (a), (b), (c), and (d) represent the CC, DHJ, DM2, and SS sites, respectively. Yellow shading represents the standard deviations of Pandora NO₂ VCD, and bars show those of GEMS; STD = standard deviation.

The hourly variations of NO₂ VCD obtained from Pandora and GEMS are shown in Fig. 6 and compared for each of the four Seosan sites in Fig. 7. Figure 6 shows a good agreement between Pandora and GEMS for all time periods. Since the GEMS measures six times in winter (10:00 – 15:00), but the Pandora NO₂ VCDs were retrieved from sunrise to sunset when SZA was less than 80°, Pandora NO₂ VCDs have a slightly more widespread trend. In Fig. 7, the differences in the diurnal Pandora NO₂ VCD variations among the sites imply the inhomogeneity of the spatial tropospheric NO₂ columns over the sites. The hourly characteristics observed at the DHJ site could possibly be affected by emissions from the petrochemical complex located approximately 16 km northwest of the site (see Fig. 1). There appears to be a discrepancy in the NO₂ peaks observed from Pandora and GEMS at the CC site, where GEMS shows enhanced NO₂ columns at 12:00 and 14:00 LT. The NO₂ columns observed from GEMS show hourly patterns similar to those from Pandora at the DHJ site. At the DM2 site, the Pandora and GEMS VCD patterns were consistent, with both displaying peaks at 11:00 LT, followed by a decreasing trend. Overall, the NO₂ VCD from Pandora and GEMS showed hourly variations, although those from Pandora tended to have slightly higher

values than those from GEMS. There could be several reasons for this difference, which are discussed later. Further quantitative comparisons of the Pandora and GEMS data were performed, as discussed below.

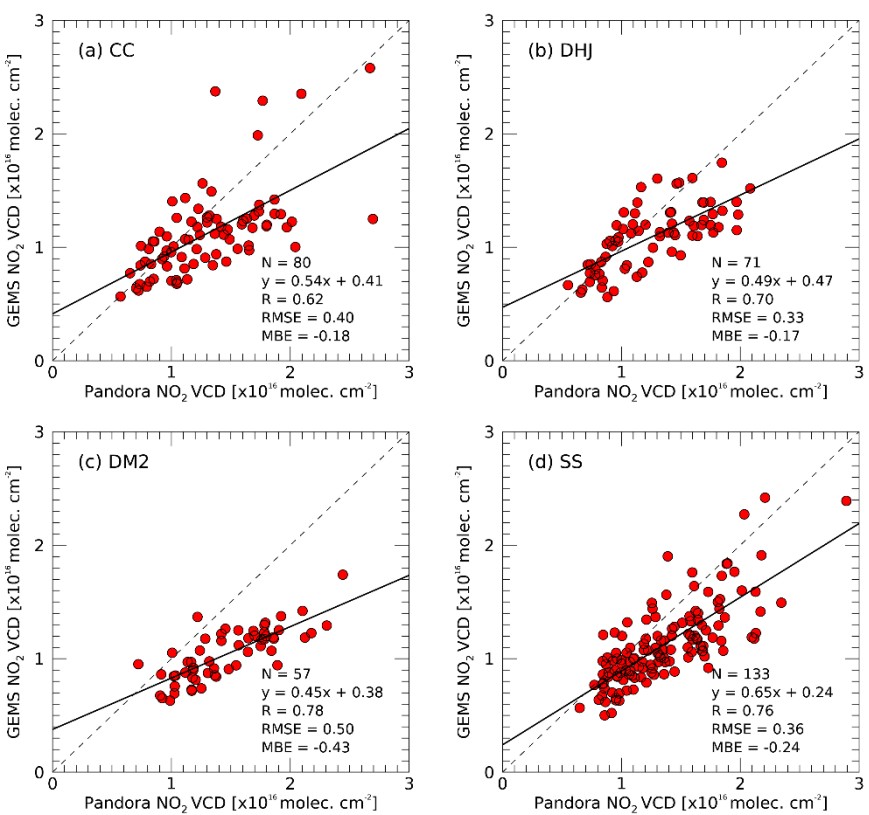

**Figure 8.** The scatterplot of NO₂ VCD between Pandora and GEMS in the CF < 0.3. (a), (b), (c), and (d) represent the CC, DHJ, DM2, and
SS sites, respectively. The grey dashed line represents the 1:1 line and the black solid line represents the regression line.

Figure 8 shows the correlations between the NO₂ VCD for the Pandora and GEMS measurements at the four Seosan sites with CF < 0.3. The R values range from 0.62 and 0.78, with values of 0.62, 0.70, 0.78, and 0.76 at the CC, DHJ, DM2, and SS sites and slopes of 0.54, 0.49, 0.45, and 0.65, respectively. Although these comparisons were conducted over a short period, the
NO₂ VCD retrieved from the geostationary GEMS measurements showed good correlations with those observed from ground-based Pandora measurement sites. The root mean square errors (RMSE) of the GEMS NO₂ against Pandora were 0.40, 0.33, 0.50, and 0.36 at the CC, DHJ, DM2, and SS sites, respectively, while the mean bias errors were -0.18, -0.17, -0.43 and -0.24, respectively.

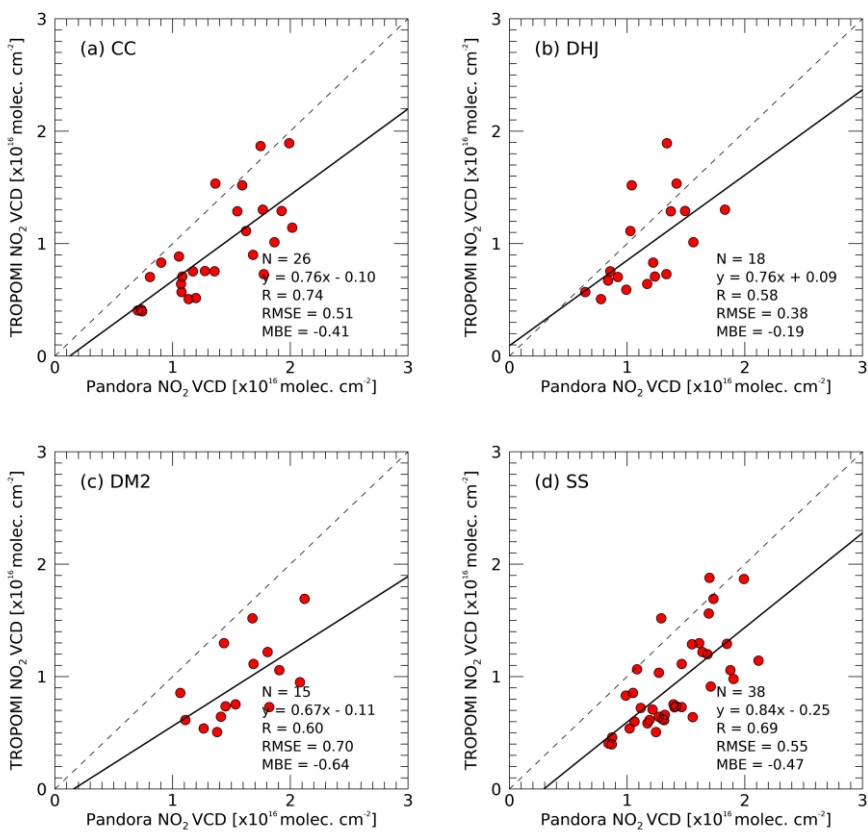

**Figure 9.** The scatterplot of $NO_2$ VCD between Pandora and TROPOMI. (a), (b), (c), and (d) represent the CC, DHJ, DM2, and SS site, respectively. The grey dashed line represents the 1:1 line and the black solid line represents the regression line.

Since GEMS is the first GEO satellite and differs from the LEO satellite with observation geometry, an additional comparison was conducted with the LEO satellite TROPOMI. TROPOMI $NO_2$ total columns used for comparison with Pandora $NO_2$ and downloaded from Copernicus open data access hub (https://s5phub.copernicus.eu; last access: 07 January 2021). TROPOMI offline channel (OFFL) dataset data were used with a quality assurance (QA) value larger than 0.75 and a cloud radiance fraction less than 0.3. In the same way as comparing Pandora and GEMS, pixels close to the Pandora measurement sites were selected and compared. The correlation coefficients between $NO_2$ total column from Pandora and TROPOMI are shown in Fig. 9 and range from 0.58 to 0.74. For the CC, DHJ, DM2, and SS sites, RMSE of the TROPOMI $NO_2$ against Pandora is calculated to be 0.51, 0.38, 0.70, and 0.52 and MBE were -0.42, -0.19, -0.64, and -0.46, respectively. In the case of GEMS, the RMSE was slightly smaller than that of TROPOMI, and there was a tendency toward underestimation less.

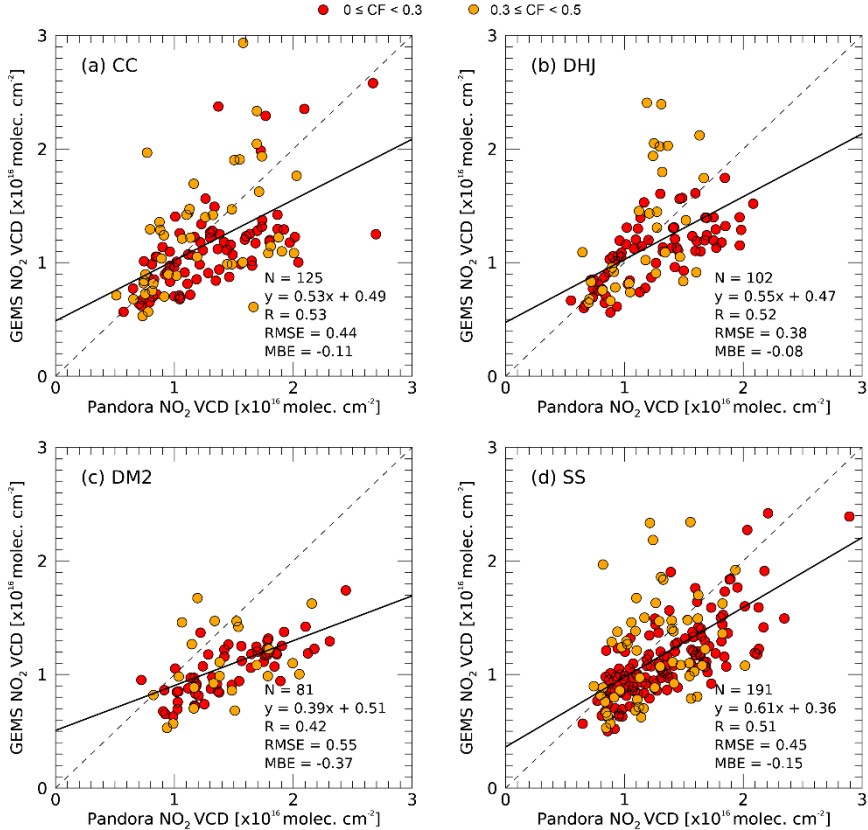

**Figure 10.** The scatterplot of $NO_2$ VCD between Pandora and GEMS in the CF conditions < 0.5 (a), (b), (c), and (d) represent the CC, DHJ, DM2, and SS sites, respectively. The colored dots mean different ranges of CF. The grey dashed line represents the 1:1 line and the black solid line represents the regression line.

Figure 10 and 11 shows the correlations between the $NO_2$ VCD obtained from the Pandora and GEMS measurements with the CF < 0.5 and < 0.7, respectively. R values tend to decrease with the increasing CF value and are in the ranges of 0.42–0.53 for CF < 0.5 and 0.35–0.48 for CF < 0.7, with slopes of 0.53, 0.55, 0.39, and 0.61 and 0.54, 0.62, 0.38, and 0.62 at the CC, DHJ, DM2, and SS sites, respectively. The RMSE of the GEMS $NO_2$ VCD against the Pandora $NO_2$ values tended to increase with high CF value and the correlation coefficient decreased (Fig. 13). The high correlation coefficient and low RMSE in the low CF conditions indicate that the diurnal $NO_2$ variations observed by the GEMS were consistent with those of Pandora under less cloudy conditions. The tendency of the correlation coefficient and RMSE against the variations in CF conditions implies that enhanced cloud conditions may degrade the sensitivity of the GEMS measurement to $NO_2$ molecules present below or at the cloud layers. However, given the discrepancies among the $NO_2$ VCD from the four Pandora instruments at the same SS location, especially under cloudy conditions (CF > 0.3; Fig. 5), the weaker correlations between the GEMS and Pandora data under higher CF conditions may be partly due to the uncertainties in the Pandora $NO_2$ VCD at high CF.

Variations in MBE with CF can be seen in Fig. 13, showing that the negative bias of GEMS against Pandora generally decreased with increasing CF. Indeed, a positive bias was observed at the DHJ site with CF < 0.7. Except for the DM2 site, the magnitudes of the negative bias at the high CF value (< 0.7) were quite small compared with those at CF < 0.3. The increasing negative bias in GEMS NO$_2$ compared to that in Pandora could be associated with GEMS CF, which was used to calculate the GEMS NO$_2$ AMF. Regarding the Pandora NO$_2$ VCD as being closer to the true values than those of the GEMS,

the large negative bias of the GEMS at low CF implies that the GEMS might underestimate the GEMS CF value, as measurement pixels with true CFs should be small. An underestimated GEMS CF may lead to an increase in the AMF and eventually to an underestimation of the NO$_2$ VCD in the pixels. Further investigation is required to identify the relationship between the GEMS CF and the negative bias tendency of the GEMS NO$_2$ VCD under less cloudy conditions.

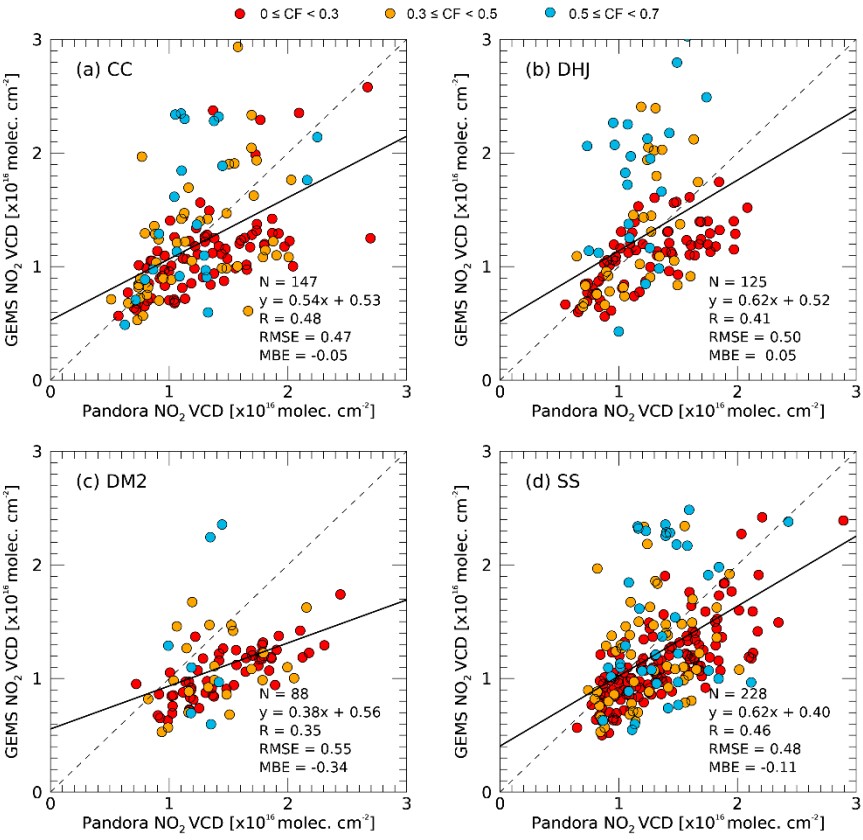

**Figure 11.** The scatterplot of NO$_2$ VCD between Pandora and GEMS in the CF conditions < 0.7. (a), (b), (c), and (d) represent the CC, DHJ, DM2, and SS sites, respectively. The colored dots mean different ranges of CF. The grey dashed line represents the 1:1 line and the black solid line represents the regression line.

## 4.3 Correction of horizontal representativeness

The GEMS pixel closest to the Pandora instrument location was used to assess the correlation between the Pandora and GEMS $NO_2$ VCD, as shown in Figs 9–11. The GEMS does not always observe the same measurement geometry, and the location of each GEMS pixel varies depending on the measurement schedule. The GEMS pixels close to the location where Pandora was installed did not completely match the Pandora observation coverage. Therefore, differences occur between spatial coverage. In particular, the $NO_2$ dSCD of Pandora was obtained from an absorption light path between the sun and the instrument at the surface. The photons from the sun reaching Pandora may pass through more than one GEMS pixel, depending on the observation geometries of the measurements. Figure 12 shows the variation in the measurement geometry of the Pandora instrument with the position of the sun. As the sun moves from east to west (morning to afternoon; (a) to (c) in Fig. 12), the direction of the viewing path of the Pandora instrument changes. The GEMS pixels corresponding to the observation path of the Pandora instrument also differ. Horizontal effects were considered using GEMS pixels and distance ratios that changed according to the observation direction, as follows: First, we selected two pixels of the GEMS, one closest to the Pandora site and another closest to the line of sight (i.e., closest to the viewing azimuth angle of the Pandora measurements). We assumed that most of the $NO_2$ was vertically distributed below 2 km altitude based on the airborne in-situ $NO_2$ measurements. The weighted mean values of GEMS $NO_2$ accounting for horizontal representativeness, were calculated as follows:

$$\text{VCD}_{\text{hr}} = \frac{d_2 \text{VCD}_1 + d_1 \text{VCD}_2}{d_1 + d_2},$$

where $\text{VCD}_{\text{hr}}$ is the $NO_2$ VCD accounting for horizontal representativeness, $d_1$ and $d_2$ are the distances between Pandora and the center of the two GEMS pixels (1 denotes the closest pixel and 2 denotes the pixel to the line of sight), and $\text{VCD}_1$ and $\text{VCD}_2$ are the GEMS $NO_2$ VCD of the two pixels.

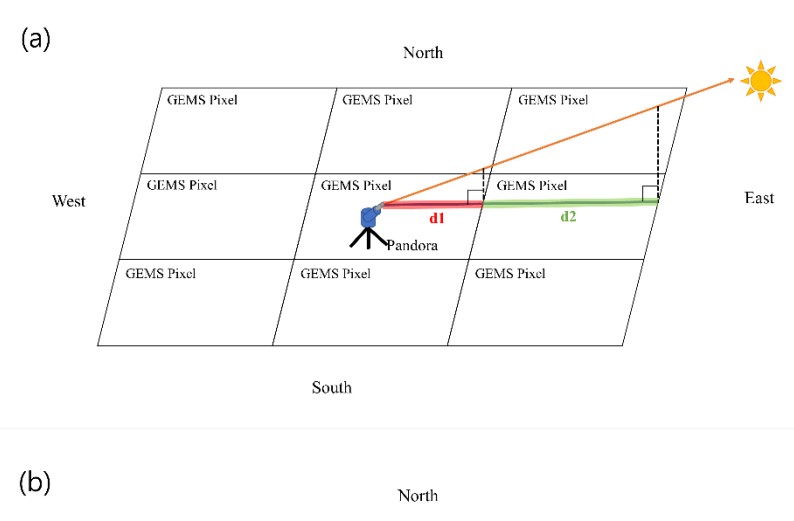

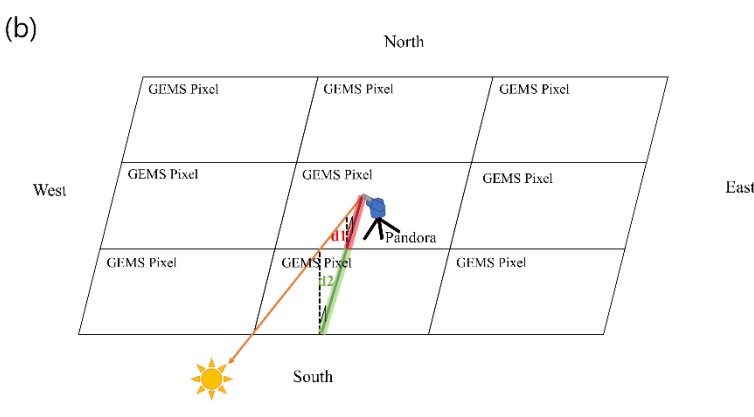

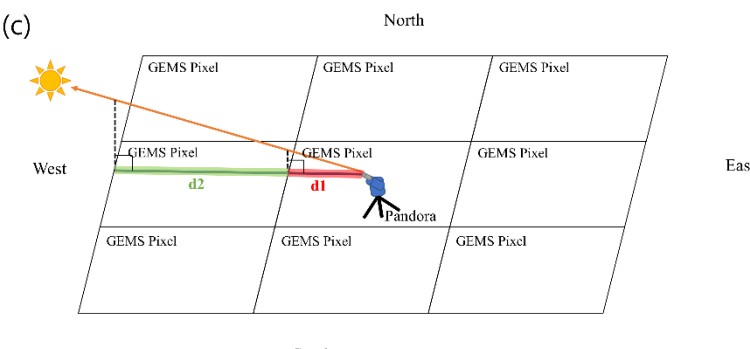

**Figure 12.** light path changes according to Pandora direct sun measurement geometry. (a), (b) and (c) represent morning, noon, and afternoon hours, respectively.

Figure 14 shows the correlations between the NO₂ VCD from Pandora and the GEMS data which were corrected for the horizontal representativeness of Pandora at CF < 0.3. The correlation coefficients were found 0.69–0.81, which were higher than those without the correction of the horizontal representativeness; the R values at the CC, DHJ, DM2, and SS sites were

0.70, 0.69, 0.81, and 0.75, respectively. Correlations at two sites CC and DM2, increased with horizontal representativeness relative to those without correction, whereas correlations at the DHJ and SS sites were similar with or without correction.

RMSEs were 0.37, 0.32, 0.49, and 0.36 with the correction, generally lower than 0.40, 0.33, 0.50, and 0.36 without the correction at the CC, DHJ, DM2, and SS sites, respectively. MBEs with the correction were similar to those without, with values of -0.18, -0.16, -0.43, and -0.2, at the CC, DHJ, DM2, and SS sites, respectively.

The viewing direction of the Pandora instrument changes depending on the location of the sun (see Fig. 12). In the case of CC, Pandora observed the downtown area from morning to noon and the rural area in the afternoon. The DM2 site observes in rural

areas in the morning and downtown areas from noon. In this case, the correlation can be improved by correcting the horizontal effect, compared to using only the nearby GEMS pixel. In contrast, the reason for the lack of significant changes in agreement before and after considering the horizontal effect in the DHJ and SS appears to be that the regional characteristics are the same according to the viewing direction. The variability of the Pandora $NO_2$ VCD with the location at a single GEMS pixel has not yet been investigated in Seosan. However, as shown by the diurnal $NO_2$ characteristics at the four sites, the $NO_2$ VCD is likely

to vary depending on the instrument location at a single GEMS pixel, causing inherent discrepancies between the GEMS and Pandora, which may partly account for the discrepancies between the horizontal and vertical measurement coverages of Pandora and GEMS. The range of statistical change was not large, but the correlation between GEMS and Pandora changed when the horizontal correction was applied to four places. Therefore, further investigations under long-term conditions and with a large number of sites are required.

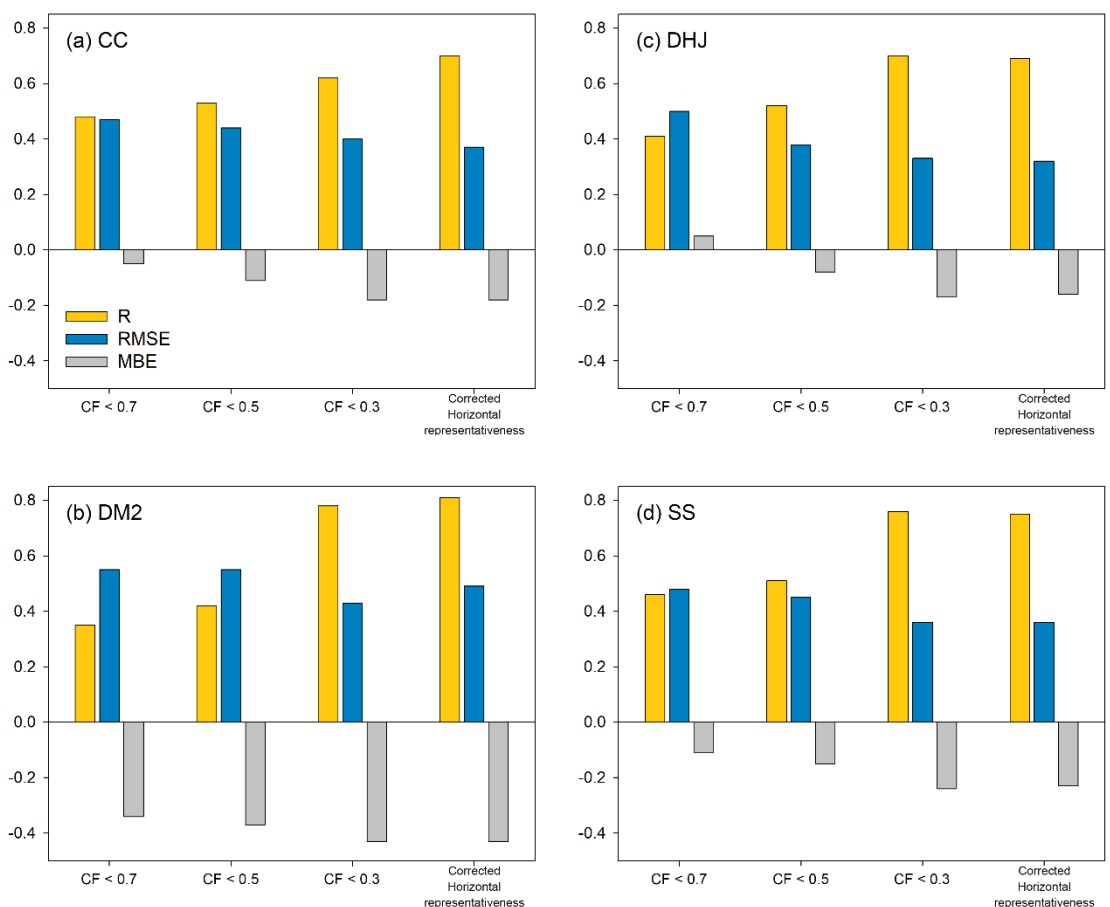

**Figure 13.** R, RMSE, and MBE between NO$_2$ VCDs obtained from Pandora and GEMS depending on the CF conditions at (a), (b), (c), and (d), which represent the CC, DHJ, DM2, and SS sites, respectively.

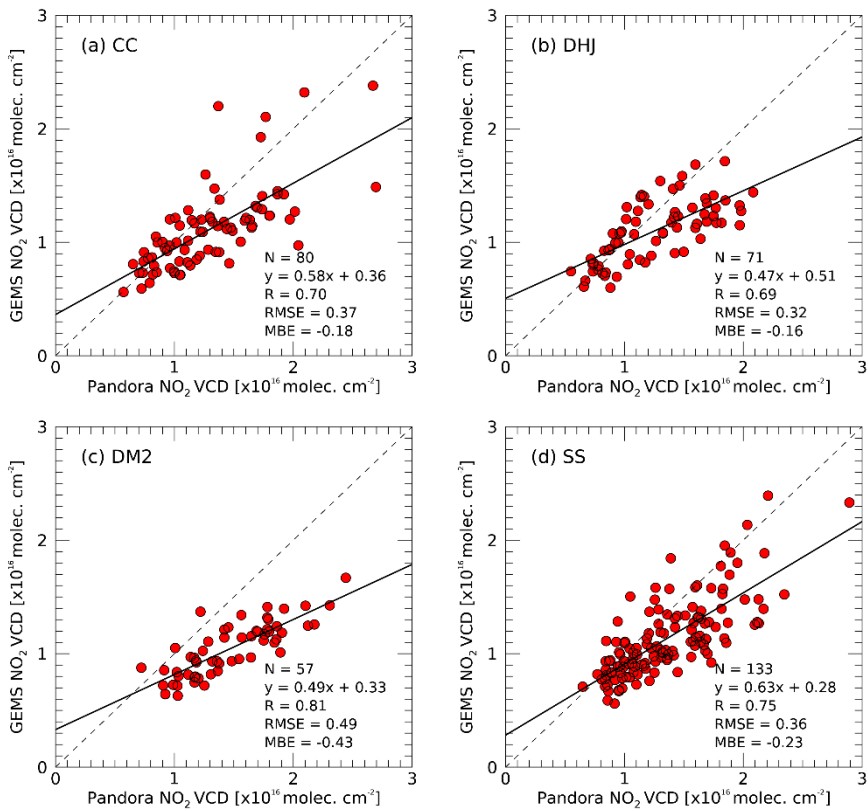

**Figure 14.** The scatterplot of NO$_2$ VCD between Pandora and GEMS with the correction for the horizontal representativeness. (a), (b), (c), and (d) represent the CC, DHJ, DM2, and SS sites, respectively. The grey dashed line represents the 1:1 line and the black solid line represents the regression line.

## 5. Conclusion

The first evaluation of GEMS NO$_2$ was conducted by comparison with NO$_2$ data obtained from ground-based Pandora measurements at four sites in Seosan, Korea. An intercomparison of NO$_2$ dVCD among the four Pandora instruments revealed a slightly decreasing agreement among instruments with increasing CF, which could partly contribute to an inherent discrepancy between the GEMS and Pandora systems at high CF. It was observed that the correlations of GEMS NO$_2$ showed good agreement with those of Pandora under less cloudy conditions (CF < 0.3). Higher correlation coefficients and lower RMSE were observed at lower CF conditions, indicating a higher sensitivity of GEMS to hourly variations in atmospheric NO$_2$ concentrations under less-cloudy conditions. The NO$_2$ VCDs may differ between GEMS and Pandora for several reasons. First, NO$_2$ cross sections at 220 K and 254.4 K were used for NO$_2$ retrieval from GEMS and Pandora, respectively. PGN

methods of $NO_2$ retrieval can lead to overestimation or underestimation depending on where tropospheric or stratospheric $NO_2$ is predominantly present (Verheolst et al., 2021). Second, there is a difference in the spatial resolution of GEMS and Pandora. However, the overall correlations or patterns between the GEMS and Pandora were very similar. We also attempted to account for the horizontal representativeness of Pandora observations. The mean correlations at the four sites increased with correction for horizontal representativeness, with maximum correlation (R = 0.81) and minimum correlation (R = 0.69) at the DM2 and DHJ sites, respectively. Variations in the correlations between sites may be attributed to variability in the $NO_2$ VCD observed by Pandora, depending on the instrument located at a single GEMS pixel. This suggests that the influence of the $NO_2$ source on the observation direction can be considered by correcting for the horizontal effect.

The first comparison of $NO_2$ VCDs from the GEMS showed relatively lower values than Pandora (MBE = -0.43–-0.17) with moderate correlations (R = -0.62–-0.78) over Seosan. $NO_2$ retrievals from the TROPOMI also showed consistent comparison results; the TROPOMI NO2 underestimated the ground-based retrievals with MBE from -0.64 to -0.19 with comparable correlations (R = 0.58–0.74). However, due to the limited Pandora measurements at the beginning of the GEMS operation, further comparisons at broader regions of GEMS FOV for long-term periods are essential for the relevant studies using the GEMS data.

*Author contributions.* DK and SK retrieved and analyzed $NO_2$ VCDs from Pandora and designed the study, while participating in the campaign. HH, LC, HL, Deok-rae K, Donghee K, JY, DL, UJ, WC, and KL planned, organized and performed the Seosan campaign. UJ, CS, SK, SP, JK, and TFH provided and supported instrument management. JK and JP provided GEMS $NO_2$ data and supported the validation process. All authors reviewed and discussed this paper.

*Financial support.* This research has been supported by the National Institute of Environmental Research of South Korea (grant no. NIER-2022-04-02-035).

*Competing interests.* The authors declare that they have no conflict of interest.

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
