# Peer review of "First-time comparison between NO2 vertical columns from GEMS and Pandora measurements"

_Atmospheric Measurement Techniques, 2023_

## Referee Comment (RC1)

Review of "First-time comparison between NO2 vertical columns from GEMS and Pandora measurements" Kim et al 2023

This paper compares NO2 column measurements from the GEMS satellite to NO2 measurements from four Pandora spectrometers in Seosan, South Korea during a three-month period (November 2020-January 2021). The four spectrometers where initially placed at the same site to evaluate their performance relative to one another. The spectrometers were then moved to separate locations near Seosan in December 2020. The effect of cloudiness on the observations was considered. Reasonable agreement between the GEMS and Pandora column NO2 was observed.

This paper fits in the scope of AMT as it presents the first validation of GEMS NO2 measurements. The overall method is fine. Most of my comments are related to clarity as parts of the text were difficult to understand. My other issue is that the manuscript does not have a strong conclusion. Numbers are given for the correlations and RMSE between Pandora and GEMS, but they are not put into context. How does the agreement between GEMS and Pandora NO2 compare to the agreement between other, similar types of measurements? Can you conclude that the GEMS data is of a quality appropriate for use in scientific studies?

Questions & Comments

Line 45: "provides diurnal variations of the NO2 VCD during daytime"

- What exactly do you mean by diurnal variations here? How frequently does the instrument take measurements over a given location?
- It would be useful to provide some information regarding why measurements of the NO2 diurnal cycle are important (ie. NOx chemistry).

Line 56: Does "diurnal NO2 VCD retrievals" mean that the retrievals have some special considerations for diurnal effects, or just that you are doing a retrieval at multiple times throughout the day?

Section 2.2 only discusses the GEMS NO2 data, however the GEMS cloud fraction is also quite important in the analysis. It would be good to discuss the CF retrieval in this section so that later results can be better understood.

Section 2.3 is quite difficult to follow. Please provide more detail on how the reference spectrum was chosen, and on how the spectral fitting was done.

Line 137: What is meant by "differences between the Pandora NO2 retrievals"? Is this referring to differences between measurements from the four Pandora instruments? Or are there multiple versions of the retrieval?

Figure 3: Maybe use different colors, in addition to different symbols, for each Pandora? The figure resolution is quite low so it is difficult to distinguish the shapes.

Line 160: Do you have an idea as to why the NO2 was higher during morning and late afternoon?

Line 161: Why did you choose 0.3 as the threshold value for "cloudiness"? The cloud fraction was also > 0.3 on November 25, but with less spread in the NO2 observations. Do you have a theory as to why? It is also interesting that the measurements only disagree for about an hour near 12:00 on November 13, even though the cloud fraction was >= 0.6 for most of the day.

Line 166-167: What do you mean by "produced least fitting errors"? Why is P1 is considered the reference spectrometer?

Line 200: The text says fig 5 shows daily variations, but the figure caption mentions hourly variations. Which is it? There also is not any discussion of Fig 5. Why is there more spread in the Pandora NO2 compared to the GEMS NO2?

Line 206: You say that there is good agreement between pandora and GEMS at the DM2 site. Is this panel c of fig 6? Because to me it looks like the agreement between Pandora and GEMS is the worst at this site. The GEMS NO2 is biased even lower than the Pandora NO2 at this site, compared to the other 3 sites.

Figure 7: Are these hourly, or daily, values?

Line 224: Is there a missing reference for a study that uses TROPOMI? Or are you comparing Pandora to TROPOMI, in addition to GEMS? Please clarify what is being referred to in this paragraph.

Figure 8: Are you including all data point with CF<0.5, or only those with 0.3<CF<0.5? I think it is the former, in which case it would be useful if the points were colored according to the CF value so that it is easier to distinguish the differences between figures 7, 8, and 9.

Line 255-256: I'm not sure the decreasing bias with increasing CF is physically meaningful... it makes sense for the mean bias to be smaller when there is more spread in the data on either side of the 1:1 line.

Figure 10: What is the CF for the 'corrected horizontal representativeness' column?

Minor Edits

In general, I suggest further proofreading as there are many grammatical errors. I have only mentioned some of the issues here.

Sentence line 27-29: wording, change to:

"With a correction for horizontal representativeness in Pandora measurement coverage, correlation coefficients ranging from 0.69 to 0.81 with RMSEs from 3.2 × 1015 molec. cm-2 to 4.9 × 1015 molec. cm-2 were achieved for CF < 0.3, showing better correlation with the correction than without the correction."

Line 32-33: The way this is currently written is unclear. When you say "plants", are you referring to N2O emissions from agricultural fertilization? Or biomass burning?

Line 48-49: Sentence is unclear. Maybe the words "by the" on line 49 should be removed?

Line 64-66: Sentence is unclear. Is the AMF more accurate for direct sun DOAS, or for MAXDOAS?

Line 67: Remove "comparison of"

Line 72-73: I do not understand this sentence. Were there two campaigns?

Line 95: manufactured with the same optics and spectrograph?

Line 101-102:

Change to: "The GEMS, a hyperspectral UV-Vis image spectrometer covers a wavelength range of 300–500 nm with a full width at half maximum (FWHM) of about 0.6 nm. GEMS measures atmospheric concentrations of species that affect air quality…"

Line 120-121: Unclear

Line 124: Should refer to Figure 2

Line 130: Is the grey line the difference between the absorption signal and the fit?

Line 149: remove "g" from "AMFg"

Line 183: Change to "comparisons were performed the closet GEMS pixels to each Pandora station"

Line 211: Sentence is unclear.

Line 219: Missing decimal point in 0.45

Line 266-268: Sentence is unclear.

---

## Author Response (AR1)

**Reply – RC1**

Thank you for your comment and advice. Accordingly, I revised the paper.

Questions & Comments

Line 45: "provides diurnal variations of the NO2 VCD during daytime"

- What exactly do you mean by diurnal variations here? How frequently does the instrument take measurements over a given location?

- It would be useful to provide some information regarding why measurements of the NO2 diurnal cycle are important (ie. NOx chemistry).

**Answer:** Since GEMS measures east-Asia area hourly during the day time, the 'diurnal variation' means hourly data can be provided. I modified 45th line sentence as follows.

- The recently launched GEMS (Geostationary Environment Monitoring Spectrometer) onboard the GEO-KOMPSAT-2B (Geostationary Korea Multi-Purpose Satellite 2B) provides hourly $NO_2$ VCD during daytime over Asia since February 2020 (Kim et al., 2020).

I agreed with your opinion and added the importance of measuring $NO_2$ diurnal cycle in Section 1(Introduction) as follows.

- The $NO_2$ diurnal cycles is important factors to understand the temporal pattern such as NOx emission, chemistry, deposition, advection, diffusion, and convection (Li et al., 2021).

Line 56: Does "diurnal $NO_2$ VCD retrievals" mean that the retrievals have some special considerations for diurnal effects, or just that you are doing a retrieval at multiple times throughout the day?

**Answer:** The latter meaning is right. I removed that word ('diurnal'), because it can be misunderstanding.

Section 2.2 only discusses the GEMS NO2 data, however the GEMS cloud fraction is also quite important in the analysis. It would be good to discuss the CF retrieval in this section so that later results can be better understood.

**Answer:** According to your advising, I added a brief about CF retrieval (at 105 line) and reference as follows.

- GEMS cloud fraction (CF) is retrieved using O2-O2 absorption properties and DOAS (Choi

et al., 2020). And we used CF for comparison of NO2 VCDs (more details can see Sect. 3).

Section 2.3 is quite difficult to follow. Please provide more detail on how the reference spectrum was chosen, and on how the spectral fitting was done.

**Answer:** I added the details on spectral fitting and reference spectrum choose method as follows.

- The DOAS method retrieve trace gases concentration by separating trace gas absorption cross section to slowly and rapidly varying part (Honninger et al., 2004).

- The reference spectrum used fitting is measured around noon on a clear day (Herman et al., 2009). It refers a spectrum with least amount of $NO_2$ presence to carry out optical density fitting during a certain period.

Line 137: What is meant by "differences between the Pandora NO2 retrievals"? Is this referring to differences between measurements from the four Pandora instruments? Or are there multiple versions of the retrieval?

**Answer:** The former meaning ('differences between measurements from the four Pandora instruments') is right. And I changed the sentence as follows.

- $NO_2$ retrievals from the four collocated Pandora instruments shows consistency of the processed data as shown in figure 4.

Figure 3: Maybe use different colors, in addition to different symbols, for each Pandora? The figure resolution is quite low so it is difficult to distinguish the shapes.

**Answer:** Yes, I changed the symbols colors in Figure 3. (Figure number changed from 3 to 4; see supplement 1)

Line 160: Do you have an idea as to why the NO2 was higher during morning and late afternoon?

**Answer:** At midday emissions are relatively low than during rush hours that has $NO_2$ emissions from vehicle (Zhao et al., 2020). As Seosan is a sub-urban area, it can be affected by the commute time. This also added on lines 163-164.

Line 161: Why did you choose 0.3 as the threshold value for "cloudiness"? The cloud fraction was

also > 0.3 on November 25, but with less spread in the NO2 observations. Do you have a theory as to why? It is also interesting that the measurements only disagree for about an hour near 12:00 on November 13, even though the cloud fraction was >= 0.6 for most of the day.

**Answer:** When I tested using multiple units, retrieved NO2 more spread noticeably around 0.3 threshold. As you said, there are cases NO2 less spread in CF > 0.3. That cases can be caused by difference in FOV (field of view) between GEMS and Pandora. The GEMS spatial resolution is 3.5 km * 8 km, and the Pandora measurement area could be clear sky even if GEMS retrieved high CF. This also added on Sect. 4.1 as follows.

- However, GEMS spatial resolution is 3.5 km * 8 km, and measurement area of Pandora could be clear sky even if GEMS retrieved high CF. So there are cases NO2 less spread in CF > 0.3 which can be caused by difference in FOV between GEMS and Pandora.

Line 166-167: What do you mean by "produced least fitting errors"? Why is P1 is considered the reference spectrometer?

**Answer:** The fitting error means retrieval error. Since P1 had minimum retrieval error and maximum retrieved data among 4 Pandora instruments, I considered as reference spectrometer.

Line 200: The text says fig 5 shows daily variations, but the figure caption mentions hourly variations. Which is it? There also is not any discussion of Fig 5. Why is there more spread in the Pandora NO2 compared to the GEMS NO2?
**Answer:** I'm sorry for confusion. 'Daily' has been changed to 'Hourly' because Figure 5 shows hourly variation. And I added the mentions with answer about your question as follows.

- Figure 5 shows good agreement between Pandora and GEMS in all time periods. Since GEMS measures 6 times in winter (10:00 – 15:00), but Pandora NO2 VCDs were retrieved from sunrise to sunset when SZA was less than 80°, Pandora NO2 VCDs has slightly more widespread trend.

Line 206: You say that there is good agreement between pandora and GEMS at the DM2 site. Is this panel c of fig 6? Because to me it looks like the agreement between Pandora and GEMS is the worst at this site. The GEMS NO2 is biased even lower than the Pandora NO2 at this site, compared to the other 3 sites.

**Answer:** That sentence (line 206) means DM2 site has most high correlation. But I removed it because it didn't suit with fig 6 that shows mean trend as you said. And the correlation is mentioned

at line 232.

Figure 7: Are these hourly, or daily, values?

**Answer:** Figure 7 shows correlation of hourly GEMS $NO_2$ data and Pandora $NO_2$ averaged within 10min from center of GMES observation time. This is mentioned in Sect. 4.2.

Line 224: Is there a missing reference for a study that uses TROPOMI? Or are you comparing Pandora to TROPOMI, in addition to GEMS? Please clarify what is being referred to in this paragraph.

**Answer:** The latter is right. I made the additionally comparison with TROPOMI. As you said, I clarify the paragraph as follows.

- An additional comparison was made with the LEO satellite TROPOMI in this study. ~

Figure 8: Are you including all data point with CF<0.5, or only those with 0.3<CF<0.5? I think it is the former, in which case it would be useful if the points were colored according to the CF value so that it is easier to distinguish the differences between figures 7, 8, and 9.

**Answer:** Yes, former is right. I changed the dot color by CF threshold. (see supplement 2)

Line 255-256: I'm not sure the decreasing bias with increasing CF is physically meaningful… it makes sense for the mean bias to be smaller when there is more spread in the data on either side of the 1:1 line.

**Answer:** I agree with you, So I removed the sentence.

Figure 10: What is the CF for the 'corrected horizontal representativeness' column?
**Answer:** In that case, CF threshold is 0.3. In Section 4.3, We described more detail about 'corrected horizontal representativeness' CF.

In general, I suggest further proofreading as there are many grammatical errors. I have only mentioned some of the issues here.

**Answer:** I edited the issues you mentioned and will correct the overall grammar errors.

Sentence line 27-29: wording, change to:

"With a correction for horizontal representativeness in Pandora measurement coverage, correlation coefficients ranging from 0.69 to 0.81 with RMSEs from $3.2 \times 10^{15}$ molec. $cm^{-2}$ to $4.9 \times 10^{15}$ molec. $cm^2$ were achieved for CF < 0.3, showing better correlation with the correction than without the correction."

**Answer:** I changed the sentence as you said.

Line 32-33: The way this is currently written is unclear. When you say "plants", are you referring to N2O emissions from agricultural fertilization? Or biomass burning?

**Answer:** It doesn't mean plant, so I changed word to "power plants"

Line 48-49: Sentence is unclear. Maybe the words "by the" on line 49 should be removed?

**Answer:** I removed "by the".

Line 64-66: Sentence is unclear. Is the AMF more accurate for direct sun DOAS, or for MAXDOAS?

**Answer:** That sentence refers to the difference of AMF calculation between direct-sun and MAX-DOAS. So, I changed the sentence as follows.

- However, uncertainties of $NO_2$ VCD retrievals by the AMF calculation is low as it utilizes simple geometric AMF (Herman et al., 2009).

Line 67: Remove "comparison of"

**Answer:** I removed "comparison of".

Line 72-73: I do not understand this sentence. Were there two campaigns?

**Answer:** The sentence refers to another campaign conducted 2021, but we didn't cover it and removed sentence to avoid confusion.

Line 95: manufactured with the same optics and spectrograph?

**Answer:** Yes, I changed "by" to "with".

Line 101-102:

Change to: "The GEMS, a hyperspectral UV-Vis image spectrometer covers a wavelength range of 300–500 nm with a full width at half maximum (FWHM) of about 0.6 nm. GEMS measures atmospheric concentrations of species that affect air quality…"

**Answer:** I changed the sentence as you said.

Line 120-121: Unclear

**Answer:** I changed the sentences.

-   . $NO_2$ differential slant column density (dSCD) was obtained using the absorption cross-sections for $NO_2$ 254.5K calculated using 220K and 294K (Vandaele et al., 1998) and $O_3$ 225K (Serdyuchenko et al., 2014), as a fourth-order polynomial in fitting window of 400-440 nm.

Line 124: Should refer to Figure 2

**Answer:** I referred figure 2.

Line 130: Is the grey line the difference between the absorption signal and the fit?

**Answer:** Yes, It's right.

Line 149: remove "g" from "AMFg"

**Answer:** I changed "AMFg" to "AMF".

Line 183: Change to "comparisons were performed the closet GEMS pixels to each Pandora station"

**Answer:** I changed sentence as you said.

Line 211: Sentence is unclear.

**Answer:** I removed the sentence because it was unclear and had the same meaning as the previous sentence.

Line 219: Missing decimal point in 0.45

**Answer:** I added the dot. Thank you.

Line 266-268: Sentence is unclear.

**Answer:** I changed the sentence as follows.

- The photons from the Sun reaching Pandora might pass more than one GEMS pixel depending on the observation geometries of the measurements, which are accounted for by the following procedure.

**Reply – RC2**

Thank you for your comment and advice. Accordingly, I revised the paper.

My main concern about this manuscript cannot be solved – it shows validation based on a short time series of observations in just one local area, and, therefore, has only limited value. Nevertheless, it is the first such a comparison for GEMS, and therefore deserves publication.

My second major concern is that the computation of Pandora vertical columns is unclear. When comparing the Pandora instruments, differential VCDs are used, defined as the difference in vertical column at a given time and the vertical column at the time of reference measurement. This is fine for the intercomparison, but not for GEMS validation. Conversion to full vertical columns is needed for validation and needs to be explained in the manuscript.

**Answer**: As you said, I only used $NO_2$ dVCDs when comparing the pandora instruments and I agree that dVCDs is not suitable for GEMS validation. Actually, in this study, $NO_2$ VCD was used for comparison with GEMS. I added the explanation about this part and conversion of full vertical columns in Sect 3.

- During the intercomparison, since clear days were not sufficient to calculate background concentration, I compared Pandora instruments using dVCD. $NO_2$ VCDs were used on comparison with GEMS.

My third major comment is that the validation, in spite of providing many numbers, remains qualitative as no attempt is made to link the results in any way to the algorithms and assumptions used in the GEMS data processor.

**Answer**: I added the sentences on Sect 5 as follows.

- NO2 cross sections 220 K and 254.4 K were used for NO2 retrieval from GEMS and Pandora, respectively. PGN methods of NO2 retrieval could lead to overestimated or underestimated depending on where tropospheric or stratospheric NO2 is predominantly present (Verheolst et al., 2021). Additionally, there is a difference in the spatial resolution of GEMS and Pandora. These reasons may cause differences in NO2 VCD values between GEMS and Pandora. However, correlations or patterns between GEMS and Pandora were very similar.

The manuscript needs more proofreading. In its current form, it contains many small mistakes and is in parts hard to understand.

**Answer**: I will supplement additionally and correct the manuscript.

Detailed comments

Title: Check author names, at least Chang-Keun Song appears to be misspelled

**Answer**: I rechecked the author names, and they are all correct.

Somewhere at the beginning of the paper you need to point out you are only looking at total columns, not tropospheric columns.

**Answer**: Yes, as you said, there may be confusion. Therefore, I pointed out that only total columns were used.

Line 66: Raman scattering is not the relevant point here. MAX-DOAS instruments used scattered light, and therefore AMFs need to be based on RTM calculations taking multiple scattering into account.

**Answer**: I agree with you. And since MAX-DOAS is less relevant to this study and line 66 could be confusing, I corrected the sentences as follows.

- However, uncertainties of $NO_2$ VCD retrievals by the AMF calculation is low as it utilizes simple geometric AMF (Herman et al., 2009).

Figure 2 legend and elsewhere: The term "deconvolution" for the separation of the optical depth into the different contributions is unusual. Please rephrase.

**Answer**: Yes, I changed the "deconvolution" to "fitted slant column optical depths".

- Fitted slant column optical depths example for November 28 2020 at 10:43:37 LT for P1. The black line represents the absorption signal, and grey line represents the absorption signal and fit residual.

Figure 3 and elsewhere: Why did you use the GEMS cloud fraction and not the Pandora intensity for this selection? Clouds can vary over scales much smaller than 4 x 8 km2.

**Answer**: The reason I used GEMS cloud fraction is Pandora data can be affected high clouds, and I wanted to find out when Pandora VCDs have discrepancy depending on GEMS CF. But as you said, Clouds can vary over scales. I tried retrieval of cloud using Pandora intensities using SMART-s algorithm (Jeong et al., 2020). In that case, more specific clouds information can be check than GEMS CF. So, I added the figure (can see supplement 3).

Line 143: Increasing the integration time does not change the ratio between direct sun light and scattered light. The problem is not the longer integration time but the larger relative contribution of scattered light in the presence of clouds .

**Answer**: Yes, I agree with you and correct as follows.

- However, in cloudy conditions, all Pandora instruments may not see the same location of Sun due to an inhomogeneity of cloud thinness. In thick cloudy conditions compared with those of clear sky, it may lead to the inclusion of unwanted stray light and increase detector noise by cloud scattering.

Line 159: VCD or dVCD?

**Answer**: dVCD is right, and I corrected it.

Figure 4: In my opinion, it would make more sense to show all data in the top row of figures and only the good data in the bottom row (cf < 0.3)

**Answer**: I had plotted the figure as you mentioned (CF < 0.3). But It was not enough to show the difference between conditions (CF < 0.3) and entire data because the correlation or slope was similar. On the other hands, the cases of CF > 0.3 better show the difference with entire data than former. So, I chose the figures of condition CF > 0.3.

Line 184: Please explain how you computed the VCD

**Answer**: I explained conversion of VCD as follows, "$NO_2$ VCD is obtained by dividing the $NO_2$ SCDs

by geometric AMFs." at line 125.

Figure 6: Is this all data for GEMS and Pandora or only matching data pairs?

**Answer**: I used only matching data pairs and added it on caption of Figure 6, as follows.

- Hourly mean NO2 VCD using only matched data from Pandora (orange line) and GEMS (black solid circles).

Figure 6: Please add station acronym to the sub-plots

**Answer**: Yes I added it.

Figure 6: Please add time periods shown to figure caption

**Answer**: Yes I added it.

Line 208: I do not think that the hourly variations are negligible – maybe they are not significant in the averages, but I assume that there are large variations in the data over time

**Answer**: "Negligible" on line 208 means that does not indicate a significant trend. But as you said, word ("Negligible") can cause confusion. So, I changed sentences as follows.

- Overall, $NO_2$ VCD from Pandora and GEMS show hourly variations, although those of Pandora tended to have slightly higher values than those of GEMS.

Line 230: Please add a figure showing the validation results for TROPOMI.

**Answer**: Yes I added it (can see supplement 4).

Line 266: Scattering by NO2 has no effect (nearly all scattering is on N2, some on O2)

**Answer**: Since Line 266 can make confusion, I revised as follows.

- The photons from the sun reaching Pandora might pass more than one GEMS pixel depending on the observation geometries of the measurements, which are accounted for

by the following procedure.

Line 275: The method used for accounting for the spatial averaging is not fully clear to me. Maybe you could add an example figure showing the GEMS pixels, the location of the Pandora, and which of the pixels are then used for validation. With an assumed layer height of 2 km and an SZA of 45°, the averaging distance of a Pandora measurement is only 2km, but at lower sun, this will of course increase.

**Answer**: The figure was added for easy understanding of the consideration horizontal effects it (can see supplement 4).

Line 292: The improvement is not large, and also not consistent between stations.

**Answer**: The correlation between GEMS and Pandora had changes when the horizontal correction was applied at four sites. The changes are not large. But it is necessary because as mentioned in the paper, the correction of horizontal representativeness may partly account for discrepancies between horizontal and vertical measurement coverages of Pandora and GEMS. Therefore, I revised sentences as follows.

- The range of statistical change was not large, but the correlation between GEMS NO2 VCDs and Pandora NO2 VCDs changed when horizontal correction was applied to four sites. Therefore, further investigation is required under conditions of long-term and large number of sites.

**Supplement**

- Supplement 1

[Figure]

Figure 4. Time series of Pandora retrievals during the intercomparison. Circle (red), square (orange), triangle (green) and diamond (blue) symbols represent total NO2 dVCD for P1, P2, P3, and P4, respectively. Grey shade represents the GEMS cloud fraction.

- Supplement 2

[Figure]

Figure 11. The scatterplot of NO2 VCD between Pandora and GEMS in the CF conditions < 0.7. (a), (b), (c) and (d) represent the CC, DHJ, DM2, and SS sites, respectively. The grey dashed line represents the 1:1 line and the black solid line represents the regression line.

[Figure]

Figure 3. Time series of Pandora retrievals during the intercomparison. Circle (red), square (orange), triangle (green) and diamond (blue) symbols represent total NO2 dVCD for P1, P2, P3, and P4, respectively. Grey shade represents Pandora aerosol cloud thickness.

- Supplement 4

[Figure]

Figure 9. The scatterplot of NO2 VCD between Pandora and TROPOMI. (a), (b), (c) and (d) represent the CC, DHJ, DM2, and SS site, respectively. The grey dashed line represents the 1:1 line and the black solid line represents the regression line.

[Figure]

Figure 12. light path changes according to Pandora direct sun measurement geometry. (a), (b) and (c) represents morning, noon, and afternoon hours, respectively.

---

## Referee Report (RR1)

Review of "First-time comparison between NO2 vertical columns from GEMS and Pandora measurements" Kim et al 2023

Overall, this version of the manuscript is much easier to read than the previous iteration.

Comments

- Paragraph at line 246: Provide more details on the TROPOMI comparison. What is the motivation for doing this? Were the TROPOMI columns matched to the Pandora locations using the same method as is used for GEMS?
- The conclusion is still lacking some details. In particular, you mention that this is the first time that GEMS, TROPOMI, and Pandora NO2 were compared, but do not provide any details on the comparison or what the results mean. It would also be useful to discuss how the agreement between GEMS and Pandora NO2 compares to the agreement between other, similar types of measurements. Can you conclude that the GEMS data is of a quality appropriate for use in scientific studies?

Minor Edits

Line 118: November 28$^{th}$ of what year?

Line 119: change "day 28th" to "November 29th"

Figure 2: Not all panels have y-axis labelled.

Line 217: Change "fir" to "for"

Line 251: change to "less underestimation"

Figures 10 and 11: Mention dot colour meanings in the caption. Which colour is the regression line fit to?

Figure 11: I suggest using a different colour for either the green or red dots as it is difficult for colourblind people to distinguish between these.

Line 337 (equation): Are the indices on VCD_1 and VCD_2 mixed up?

Line 354: Change "GENS" to "GEMS"

---

## Author Response (AR2)

Comments

- Paragraph at line 246: Provide more details on the TROPOMI comparison. What is the motivation for doing this? Were the TROPOMI columns matched to the Pandora locations using the same method as is used for GEMS?
  - ➔ As GEMS is GEO satellite, it needs to compare with other LEO satellites. Therefore I added and revised the sentences as follows.
    - • Since GEMS is the first GEO satellite and differs from the LEO satellite with observation geometry, an additional comparison was conducted with the LEO satellite TROPOMI. TROPOMI $NO_2$ total columns used for comparison with Pandora $NO_2$ and downloaded from Copernicus open data access hub (https://s5phub.copernicus.eu; last access: 07 January 2021). TROPOMI offline channel (OFFL) dataset data were used with a quality assurance (QA) value larger than 0.75 and a cloud radiance fraction less than 0.3. In the same way as comparing Pandora and GEMS, pixels close to the Pandora measurement sites were selected and compared.

- The conclusion is still lacking some details. In particular, you mention that this is the first time that GEMS, TROPOMI, and Pandora NO2 were compared, but do not provide any details on the comparison or what the results mean. It would also be useful to discuss how the agreement between GEMS and Pandora NO2 compares to the agreement between other, similar types of measurements. Can you conclude that the GEMS data is of a quality appropriate for use in scientific studies?
  - ➔ Thank you for your opinion. And I added the details about comparison results, and evaluation as follows.
    - • The first comparison of $NO_2$ VCDs from the GEMS showed relatively lower values than Pandora (MBE = -0.43–-0.17) with moderate correlations (R = -0.62–-0.78) over Seosan. $NO_2$ retrievals from the TROPOMI also showed consistent comparison results; the TROPOMI $NO_2$ underestimated the ground-based retrievals with MBE from -0.64 to -0.19 with comparable correlations (R = 0.58–0.74). However, due to the limited Pandora measurements at the beginning of the GEMS operation, further comparisons at broader regions of GEMS FOV for long-term periods are essential for the relevant studies using the GEMS data.

Minor Edits

Line 118: November 28[th] of what year?

➔ It means November 28th, 2020. So, I added the year.

Line 119: change "day 28th" to "November 29th"

➔ I changed day 28th to November 28[th]

Figure 2: Not all panels have y-axis labelled.

➔ All panels have the same y-axis.

Line 217: Change "fir" to "for"

➔ I corrected it.

Line 251: change to "less underestimation"

➔ I changed "underestimation less" to "less underestimation"

Figures 10 and 11: Mention dot colour meanings in the caption. Which colour is the regression line fit to?

➔ I added the mention of color meaning as follows.

- Figure 11. The scatterplot of NO2 VCD between Pandora and GEMS in the CF conditions < 0.7. (a), (b), (c), and (d) represent the CC, DHJ, DM2, and SS sites, respectively. The colored dots mean different ranges of CF. The grey dashed line represents the 1:1 line and the black solid line represents the regression line.

Figure 11: I suggest using a different colour for either the green or red dots as it is difficult for colourblind people to distinguish between these.

➔ Thank you, I changed the dot color.

Line 337 (equation): Are the indices on VCD_1 and VCD_2 mixed up?

➔ The 337 line doesn't have any equation or VCD. If you mean "Intercomparison VCD (338 line)", that is not indices VCD_1 and VCD_2 mixed up. To prevent misunderstanding, I modified it to dVCD. And, when VCD_1 and VCD_2 are considered together, it is only when the horizontal effect is considered.

Line 354: Change "GENS" to "GEMS"

➔ I corrected it.